# Impact of the COVID-19 Pandemic Lockdown on Air Pollution in 20 Major Cities around the World

**Franck Fu \*, Kathleen L. Purvis-Roberts**  **and Branwen Williams**

W.M. Keck Science Department, Claremont McKenna College, Pitzer College and Scripps College, Claremont, CA 91711, USA; KPurvis@kecksci.claremont.edu (K.L.P.-R.); BWilliams@kecksci.claremont.edu (B.W.)

\* Correspondence: FFu@kecksci.claremont.edu

**Abstract:** In order to fight against the spread of COVID-19, the most hard-hit countries in the spring of 2020 implemented different lockdown strategies. To assess the impact of the COVID-19 pandemic lockdown on air quality worldwide, Air Quality Index (AQI) data was used to estimate the change in air quality in 20 major cities on six continents. Our results show significant declines of AQI in $NO_2$, $SO_2$, CO, $PM_{2.5}$ and $PM_{10}$ in most cities, mainly due to the reduction of transportation, industry and commercial activities during lockdown. This work shows the reduction of primary pollutants, especially $NO_2$, is mainly due to lockdown policies. However, preexisting local environmental policy regulations also contributed to declining $NO_2$, $SO_2$ and $PM_{2.5}$ emissions, especially in Asian countries. In addition, higher rainfall during the lockdown period could cause decline of $PM_{2.5}$, especially in Johannesburg. By contrast, the changes of AQI in ground-level $O_3$ were not significant in most of cities, as meteorological variability and ratio of $VOC/NO_x$ are key factors in ground-level $O_3$ formation.

**Keywords:** COVID-19; AQI; lockdown policy; major cities; $NO_2$; $PM_{2.5}$; ozone

## 1. Introduction

The majority of the world's major cities suffer from serious air pollution issues, leading to more than two million deaths globally through damage to the lungs and the respiratory system [1]. The International Agency for Research on Cancer (IARC) has classified air pollution as Carcinogenic to Humans (Group 1), as studies show exposure to outdoor air pollution causes lung cancer [2]. The common pollutants of concern are nitrogen dioxide ($NO_2$), sulfur dioxide ($SO_2$), carbon monoxide (CO), ground-level ozone ($O_3$), and particulate matter ($PM_{2.5}$ and $PM_{10}$). $NO_2$, $SO_2$, PM, and $O_3$ are all associated with the development and/or aggravation of respiratory diseases that reduce lung function, particularly in vulnerable populations with pulmonary disease or asthma [3–5]. CO can cause subtle cardiovascular and neurobehavioral effects even at low concentrations [6]. Because of these severe health issues associated with air pollution, these six criteria pollutants are routinely measured in many countries. If the concentration of these air pollutants is high enough to impact human heath, this can lead to governmental action plans and policies to control pollutant discharge and improve the air quality.

Air pollution is released from a wide variety of natural and anthropogenic activities [7], most pollutants are both primary and secondary. $NO_2$, $SO_2$ and CO are mainly primary and originated from anthropogenic activities globally [8–10]. Among anthropogenic sources, transportation and combustion in power plants are the primary and secondary sources of $NO_2$ [11–13] and CO [11,14]. However, the main source of $SO_2$ is generally combustion of coal in power plants and the manufacturing industries [11,15]. Ground-level $O_3$ is a secondary pollutant formed in the

air by a series of photochemical reactions, the key factors are sunlight, $NO_x$ and a variety of volatile organic compounds (VOCs). Primary particulate matter is directly released into the atmosphere by natural and anthropogenic activities, while secondary particles are formed in the atmosphere from other precursor pollutants, such as $SO_2$, $NO_2$, $NH_3$ and VOCs. In urban areas, anthropogenic sources of $PM_{10}$ (aerodynamic diameter $\leq$ 10 μm) and $PM_{2.5}$ (aerodynamic diameter $\leq$ 2.5 μm) dominate, and the major sources are residential combustion, large-scale combustion (i.e., power plants), industrial processes, agriculture and transportation (road and non-road) [16].

In December 2019, cases of pneumonia of "unknown etiology" were first identified in Wuhan, China [17]. On 11 February 2020, the World Health Organization (WHO) announced an official name coronavirus disease 2019 (COVID-19) for this epidemic disease. After the first outbreak in Wuhan, several community outbreaks occurred in February in countries outside of China, such as South Korea, Italy, Germany and Spain. In early 2020, the epicenter moved from Asia to Europe and then to the Americas by March. In order to mitigate the infection rate of COVID-19, many countries applied different lockdown strategies. Lockdown measures included partial or full closure of international borders, schools, non-essential business and citizen mobility restriction [18]. The reduction of transportation, commercial and industrial activities due to these lockdown strategies has the potential to reduce the emissions of primary pollutants and change the formation rate of secondary pollutants, due to the change of precursor emissions.

Recent studies suggested that lockdown measures contributed to improvements in air quality, especially in urban areas, where more anthropogenic activities are present (see the summary of recent studies on COVID-19 and air quality impacts [19]). Most studies assessed the impacts of COVID-19 on air quality within a single country. Few studies expanded world-wide, but in one study, remarkable declines of AQI were observed in $NO_2$ (−44% to −13%), ozone (−20% to −2%) and $PM_{2.5}$ (−28% to 10%) during the first two weeks of the lockdown in 27 countries [20]. Another focused on the global change of $NO_2$, CO and AOD (Aerosol Optical Depth), they observed reduction of $NO_2$ (0.00002 mol $m^{-2}$), CO (<0.03 mol $m^{-2}$) and AOD (~0.1–0.2) in the major hotspots of COVID-19 outbreak between February and March relative to 2019 [21]. Thus, in order to provide a more comprehensive analysis of the impact of lockdowns on all 6 critical air pollutants during the entire lockdown period and to assess the impact of different lockdown strategies on air pollution, AQI in 20 major cities worldwide was examined. Major cities were selected because they often have air pollution issues and more available air quality monitoring stations.

## 2. Materials and Methods

### 2.1. City Selection

Twenty major cities were selected (Figure 1) representing all continents except Antarctica, including seventeen cities in the most hard-hit countries: Wuhan, Beijing (China), Delhi (India), Tehran (Iran), Istanbul (Turkey) in Asia; Rome (Italy), Madrid (Spain), Paris (France), London (UK), Berlin (Germany) and Moscow (Russia) in Europe; Johannesburg (South Africa) in Africa and Los Angeles, New York City (USA), Mexico city (Mexico), Sao Paulo (Brazil) and Lima (Peru) in North and South America.

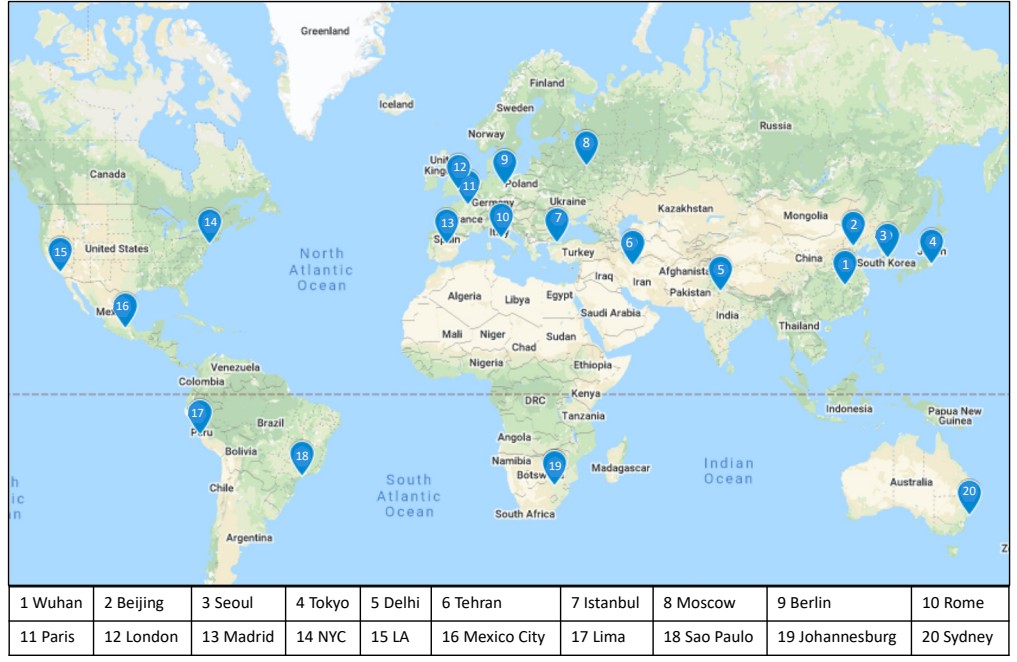

| 1 Wuhan | 2 Beijing | 3 Seoul | 4 Tokyo | 5 Delhi | 6 Tehran | 7 Istanbul | 8 Moscow | 9 Berlin | 10 Rome |
|---------|-----------|---------|---------|---------|----------|------------|----------|----------|---------|
| 11 Paris | 12 London | 13 Madrid | 14 NYC | 15 LA | 16 Mexico City | 17 Lima | 18 Sao Paulo | 19 Johannesburg | 20 Sydney |

**Figure 1.** Twenty major cities impacted by COVID-19 that applied lockdown or social distancing policies around the world.

## 2.2. Air Quality Data

After the outbreak of COVID-19, the World's Air Pollution, Real-time Air Quality Index (WAQI project) started to provide a new dedicated, dataset specific for COVID-19 related research, covering approximately 380 major cities throughout the world (aqicn.org/data-platform/COVID19/). The World Air Quality project is a non-profit started in 2007 to provide air quality information for more than 100 countries, covering more than 30,000 stations (local governmental/professional monitoring network) in 1000 major cities. The data for each major city is based on the median of several stations. The dataset provides a statistical summary for each of the air pollutant species, all air pollutants are converted to an Air Quality Index (AQI) with the U.S. Environmental Protection Agency (EPA) standard calculation. The daily median AQI was used in this study for each of 20 major cities.

The AQI is a dimensionless index that quantitatively describes air quality conditions based on standards of each pollutant, which provides a comprehensive evaluation on the combined effects of the six criteria pollutants ($NO_2$, $SO_2$, CO, ground-level $O_3$, $PM_{2.5}$ and $PM_{10}$). The AQI is classified into six grades calculated from the concentrations of various pollutants, i.e., AQI: 0–50 (Good), 51–100 (Moderate), 101–150 (Unhealthy for Sensitive Groups), 151–200 (Unhealthy), 201–300 (Very Unhealthy), and >300 (Hazardous).

## 2.3. Lockdown Data

In order to define the lockdown period for each of the cities of interest, sources including media and official government websites and research papers were used. It should be noted that it's counted from the beginning of lockdown or strict social distancing policy to the date the lockdown started to ease (Table 1), as the reopening is a complex process and could include several steps. In order to compare AQI data during the lockdown period to historical data, the same period of data for 2017, 2018 and 2019 were taken into account. The average AQI of each pollutant was compared to the previous year (2019) and to the 3-year average of the same period to calculate the decrease or increase of each pollutant.

**Table 1.** Start and end of lockdown period and lockdown policies in 20 selected major cities.

| City | Country | Lockdown | | Lockdown Policy | |
|---|---|---|---|---|---|
| | | From | To | Total/Partial | Other Local Actions |
| Wuhan | China | 23-Jan | 7-Apr | Total | Strictest human mobility restrictions, public transportation stopped |
| Delhi | India | 24-Mar | 31-May | Total | Only those who work for "essential services" can move freely, public transportation stopped |
| Lima | Peru | 15-Mar | 30-Jun | Total | Gender-based mobility restriction |
| Madrid | Spain | 14-Mar | 4-May | Total | Outdoor physical exercise banned |
| Tehran | Iran | 13-Mar | 17-Apr | Total | Shops, streets and roads cleared |
| Moscow | Russia | 30-Mar | 8-Jun | Total | Digital pass required for car or public transport use |
| Rome | Italy | 9-Mar | 3-May | Total | Only allowed to go out alone near home |
| Paris | France | 17-Mar | 10-May | Total | Permit required for going out |
| London | U.K. | 23-Mar | 10-May | Total | Only go outside to buy food, to exercise once a day, or go to work if they absolutely cannot work from home |
| Johannesburg | South Africa | 26-Mar | 30-Apr | Total | Severe restrictions on travel and movement |
| Sydney | Australia | 23-Mar | 27-Apr | Total | Stay-at-home except for essential outings |
| Beijing | China | 10-Feb | 27-Mar | Partial | Ordering residential communities and villages to limit access for outsiders |

**Table 1.** *Cont.*

| City | Country | Lockdown | | Lockdown Policy | |
| | | From | To | Total/Partial | Other Local Actions |
| --- | --- | --- | --- | --- | --- |
| New York City | U.S.A | 22-Mar | 7-Jun | Partial | Statewide stay-at-home order |
| Los Angeles | U.S.A | 19-Mar | 7-May | Partial | Statewide stay-at-home order, social pressure for violations |
| Mexico City | Mexico | 23-Mar | 30-May | Partial | Closing gyms, museums and clubs, banning big gathering, restricting mobility to areas less affected |
| Sao Paulo | Brazil | 24-Mar | 10-May | Partial | Social distancing measures |
| Berlin | Germany | 17-Mar | 19-Apr | Partial | Rules differing across states. Only go out alone or with a person from same household. |
| Seoul | South Korea | 24-Feb | 6-May | – | No strict lockdown, social distancing applied, no movement restriction |
| Tokyo | Japan | 7-Apr | 24-May | – | State of emergency, encouraged social distancing |
| Istanbul | Turkey | 21-Mar | 10-May | – | Weekend curfew |

Note: full lockdown: national lockdown, partial lockdown: regional/statewide lockdown, –: no official lockdown, only social distancing or curfew measurement.

### 2.4. Climate Data

In order to assess the impact of weather condition on the air quality, the climate data from the Global Historical Climatology Network was analyzed. This daily (GHCN-Daily) dataset includes daily land surface observations from around the world. The GHCN-Daily was developed to meet the needs of climate analysis and monitoring studies that require data on a sub-monthly time resolution. The dataset includes observations from the World Meteorological Organization, Cooperative, and Community Collaborative Rain, Hail and Snow (CoCoRaHS) networks.

The daily rainfall and daily average temperature for the lockdown period of 2020 and the same period of 2019 were from the GHCN-Daily dataset. The average temperature was not available for Los Angeles (LA), New York City (NYC) and Sydney, so the maximum temperature was used, as the maximum temperature is important for ozone formation. For Moscow, Mexico City and Berlin,

the climate data was not complete; for instance, Moscow had no data for April and May of 2020 and Mexico City was missing 15 days of data in May. The data from the weather underground website was used instead of GHCN data for those three locations. From both sources, there is no data for rainfall for Moscow, Mexico City and Berlin. The information about the location (name or/and number of the station) can be found in Supplementary Table S1.

## 3. Results

### 3.1. Primary Pollutants: $NO_2$, $SO_2$ and CO

$NO_2$ AQI decreased for all cities during the lockdown period, relative to 2019 and the past 3 years average for the same period (Table 2, Figure 2). $NO_2$ AQI declined in all cities with the highest decrease (−60%) in Delhi and the lowest (−11.1%) in Sydney in comparison to 2019 and decreased the most (−63.3%) in Wuhan and the least in Sydney (−15.5%) relative to an average of the past three years.

**Table 2.** Percentage (%) change in AQI for $NO_2$, $SO_2$, CO, ground-level $O_3$, $PM_{2.5}$, and $PM_{10}$ during lockdown period in 2020 compared to average of 2017–2019, and to 2019 single year for the same period in 20 major cities in the world. Bold red: the change is statistically significant ($p < 0.05$) relative to 2019 or to at least one of the 3 past years, according to ANOVA and Tukey HSD tests. +: increase of AQI, −: decrease of AQI.

| Continent | City | Country | $NO_2$ To 17–19 | $NO_2$ To 19 | $SO_2$ To 17–19 | $SO_2$ To 19 | CO To 17–19 | CO To 19 | Ground-level $O_3$ To 17–19 | Ground-level $O_3$ To 19 | $PM_{2.5}$ To 17–19 | $PM_{2.5}$ To 19 | $PM_{10}$ To 17–19 | $PM_{10}$ To 19 |
|---|---|---|---|---|---|---|---|---|---|---|---|---|---|---|
| Asia | Wuhan | China | **−63.3** | **−58.3** | −28.6 | −11.2 | **−17.3** | **−13.8** | **+45.6** | **+54.2** | **−26.2** | **−23.2** | **−31.9** | **−30.4** |
| | Beijing | | **−41.8** | **−33.7** | −60.3 | −33.5 | −7.1 | **+92.4** | +10.7 | +5.1 | −15.2 | −7.2 | +0.1 | −14.8 |
| | Seoul | South Korea | **−28.0** | **−25.8** | **−28.3** | **−20.8** | **−14.5** | **−14.6** | +10.9 | +10.2 | **−21.2** | **−19.1** | −8.8 | +19.5 |
| | Tokyo | Japan | −25.8 | −19.5 | **−37.6** | **−28.1** | −3.1 | +7.3 | −5.3 | −6.7 | −22.9 | −11.4 | −24.1 | −11.0 |
| | Delhi | India | **−57.7** | **−60** | **−23.7** | **−31.7** | −22.8 | **−34.0** | +19.3 | **+36.3** | **−31.0** | **−27.6** | **−47.9** | **−45.9** |
| | Tehran * | Iran | NA | **−35.2** | NA | NA | NA | NA | NA | NA | NA | **−21.9** | NA | **−37.9** |
| | Istanbul | Turkey | −36.5 | −19.9 | **+53.8** | **+29.3** | +48.5 | +2.9 | **−8.3** | **−43.6** | −19.1 | −3.4 | **−22.4** | **−19.0** |
| Europe | Moscow † | Russia | **−35.8** | **−39.8** | **−25.5** | **−25.8** | **−18.8** | **−18.7** | NA | +6.4 | −12.7 | **−25.9** | −29.6 | **−42.5** |
| | Berlin | Germany | **−45.0** | −17.6 | NA | **+29.5** | NA | NA | +16.4 | +3.9 | **−27.4** | **−22.5** | −15.1 | −10.3 |
| | Rome | Italy | **−45.0** | **−36.4** | −8.6 | −3.8 | NA | NA | +4.4 | +2.9 | +0.1 | +8.3 | −6.6 | −1.8 |
| | Paris | France | **−47.8** | **−46.4** | −20.5 | +13.5 | NA | NA | **+24.0** | **+26.8** | −4.5 | −13 | −14.7 | −22.3 |
| | Madrid | Spain | **−56** | **−51.6** | +7.1 | **−54.8** | NA | NA | −4.9 | −9.9 | −2.4 | −1.6 | −17.2 | −19.8 |
| | London | U. K | **−39.6** | **−37.8** | +1.3 | +0.6 | **−35.0** | **−53.5** | **+47.7** | **+48.0** | −8.8 | −14 | −4.7 | −10 |
| North America | New York City # | U. S | **−33.7** | **−27.5** | NA | NA | **−22.0** | **−20.7** | +1.0 | −6.3 | **−30.8** | **−27.6** | NA | NA |
| | Los Angeles | | **−29.0** | **−24.2** | NA | NA | −36.9 | 7.2 | −11.7 | −3.5 | −20.4 | −13.7 | −18.1 | +2.0 |
| | Mexico City | Mexico | **−35.2** | **−24.9** | −14.9 | +2.2 | −1.2 | +4.0 | +7.9 | −0.1 | −3.7 | **−8.3** | −8.8 | **−15.5** |
| South America | Lima ‡ | Peru | **−63.4** | **−50.5** | −18.1 | **−35.2** | **−58.6** | **−61.8** | −28.5 | **−42.9** | **−27.1** | **−19.4** | **−42.3** | **−31.7** |
| | Sao Paulo | Brazil | **−36.0** | **−37.8** | **−27.3** | **−23.2** | **−31.8** | **−32.7** | **+33.9** | **+24.9** | −11.3 | **−18.3** | −5.4 | −12.9 |
| Africa | Johannesburg § | South Africa | NA | **−23.0** | NA | −13.9 | NA | 5.0 | NA | +9.0 | NA | **−31.3** | NA | **−33.1** |
| Oceania | Sydney | Australia | −15.0 | −11.1 | NA | NA | **−25.5** | **−24** | +5.2 | +5.6 | **−34.7** | **−29.2** | −19.7 | **−17.0** |

*: No available data in 2019, the comparison was made with 2018. †: the start date is 3 April instead of 30 March according to the data availability. ‡: the start date is 29 March instead of 15 March according to the data availability. §: no available data in 2017 and 2018, only compared to 2019. #: Manhattan area of New York City.

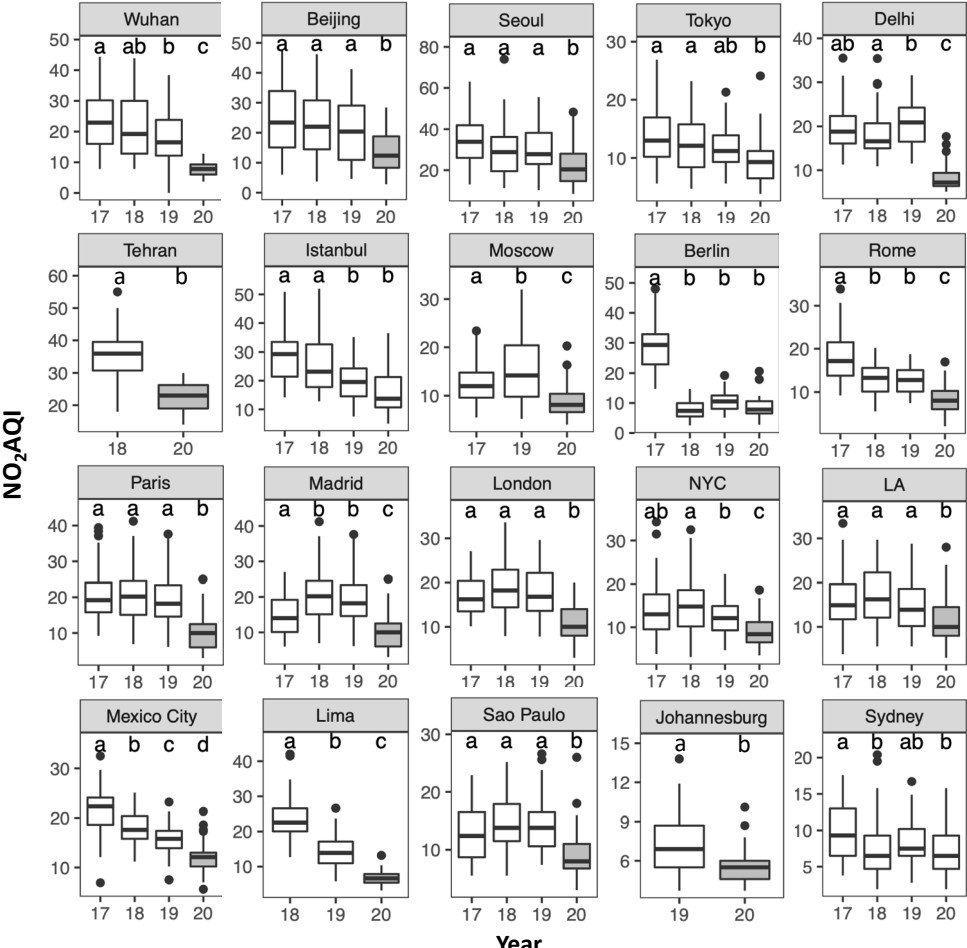

**Figure 2.** NO$_2$ AQI in 20 worldwide cities during the lockdown period in 2020 compared to the same period of 2017, 2018 and 2019. Line within the box: the median, box: first and third quartiles, whiskers: non-outlier range, dot: outliers. Years sharing the same letter mean AQI are not significantly different ($p > 0.05$). The grey box for 2020: the change in 2020 is significant relative to every previous year.

According to ANOVA and Tukey HSD tests with 95% confidence, the decreases were statistically significant for all cities, meaning the 2020 AQI was significantly different to each of past 3 years, expect for Berlin, Tokyo, Istanbul and Sydney. NO$_2$ continuously decreased from 2017 to 2020 in all Asian cities: Beijing, Wuhan, Tokyo, Istanbul and in South American cities: Lima and Mexico City (Figure 2). This trend of decreasing NO$_2$ AQI in these cities (that existed prior to 2019) contributed to the small difference between the 2017–2019 average and 2019 value. The SO$_2$ AQI decreased significantly in 6 cities, compared to each of past 3 years (Table 2, Figure S1). The highest decrease (−54.8%) occurred in Madrid relative to 2019, due to a much higher SO$_2$ level in 2019, for which the cause is unknown. However, the AQI in 2020 was not statistically different in comparison to those of 2017 and 2018, suggesting that the SO$_2$ level was stable in Madrid. In addition, significant increases of SO$_2$ AQI in Istanbul (+29.3%) and Berlin (+29.5%) were also observed (Table 2). For other cities, the changes were not statistically significant, or there was no available data.

Limited CO AQI data was available in the World Air Quality Index project data set. For cities with CO data, the CO AQI decreased significantly in eight of 15 cities (Table 2, Figure S2), with the maximum decrease in Lima (−60%) relative to 2019. In six other cities, the AQI changes were not statistically significant, compared to each of past 3 years. In Beijing, where a much higher increase of CO AQI relative to 2019 was observed (+92.4), by excluding the abnormally and unexplained low 2019 concentrations, the level of CO decreased by 26.2% relative to 2017 and 2018.

### 3.2. Secondary Pollutants: Ground-Level O₃

Contrary to the trends of decreasing primary pollutants, ground-level $O_3$ AQI increased (+2.9–+54.2%) in 12 cities and decreased (from −0.1 to −43.6%) in the other 7 (no data in Tehran) during the lockdown period for each city relative to 2019 (Table 2 and Figure 3). Comparing each of the past 3 years using an ANOVA with Tukey HSD test, the $O_3$ significantly increased in Wuhan, Paris, London and Sao Paulo, and significantly decreased in Istanbul. For other cities, the changes were not statistically significant. Wuhan experienced the maximum increase (+54.2%) and Istanbul experienced the maximum decrease (−43.6%). Lima also experienced a large decrease (−42.9%) relative to 2019, because of the dramatically higher ozone concentration in 2019 than other years (Figure 3). However, the AQI level in 2020 is statistically insignificant in comparison to 2018's level.

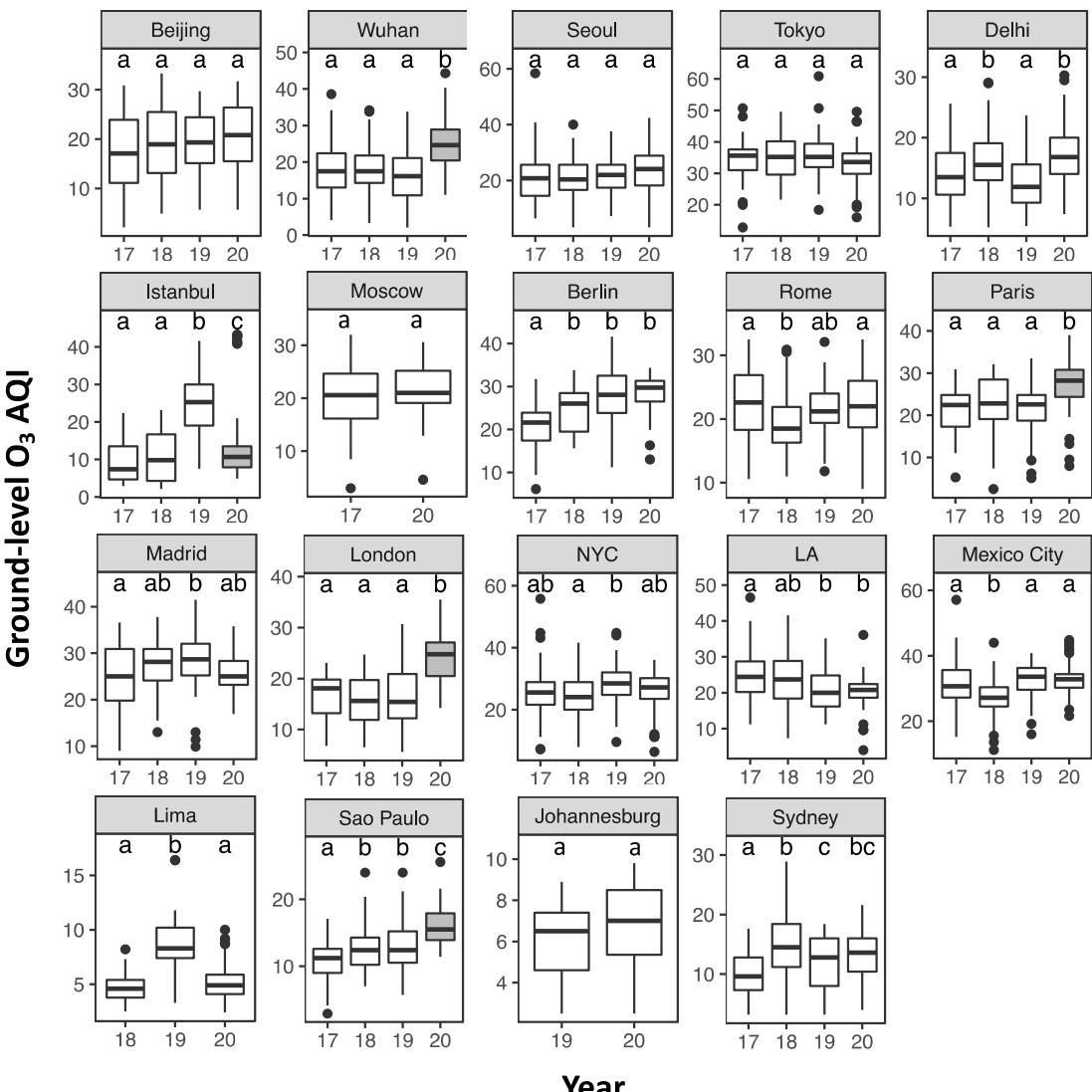

**Figure 3.** Ground-level ozone AQI in selected worldwide cities during the lockdown period in 2020 compared to the same period of 2017, 2018 and 2019. Line within the box: the median, box: first and third quartiles, whiskers: non-outlier range, dot: outliers. Years sharing the same letter mean not significantly different (*p* > 0.05). The grey box for 2020: the change in 2020 is significant relative to every previous year.

### 3.3. Particulate Matter: PM$_{2.5}$ and PM$_{10}$

PM$_{2.5}$ AQI decreased in all cities (Table 2 and Figure 4), except in Rome with an increase of +8.3% during the lockdown period relative to 2019 and the 2017–2019 reference, with the maximum decrease in Johannesburg (−31.3%) relative to 2019. The decreases are statistically significant in 12 cities relative to 2019, and in 9 cities compared to each of the past 3 years. For other cities, the changes are not statistically significant (see Table 2 and Figure 4). PM$_{10}$ also decreased in all cities, except in Seoul and Los Angeles (insignificant increases), with the maximum decrease in Delhi (−45.9%) relative to 2019. The decreases are statistically significant in 9 of 19 cities relative to 2019, and in 4 of 17 cities compared to each of the past 3 years (see Table 2, Figure 5).

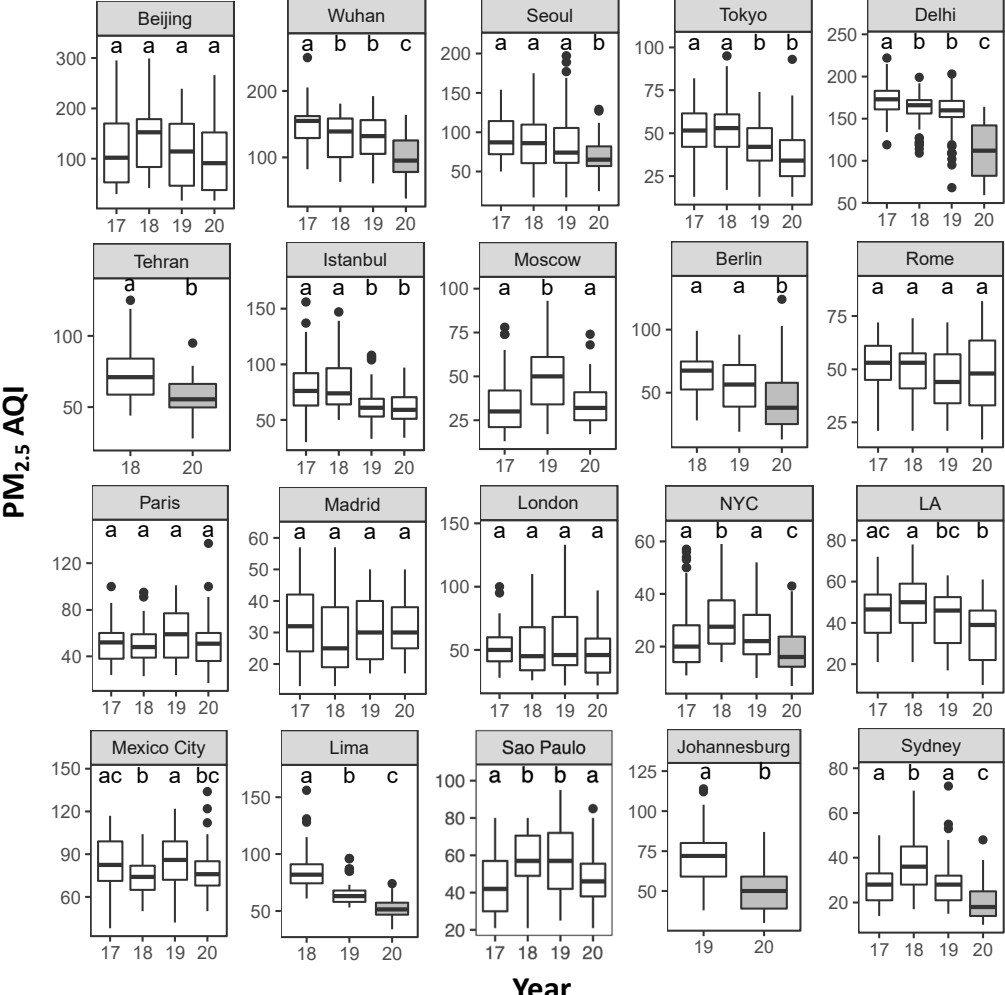

**Figure 4.** PM$_{2.5}$ AQI in selected worldwide cities during the lockdown period in 2020 compared to the same period of 2017, 2018 and 2019. Line within the box: the median, box: first and third quartiles, whiskers: non-outlier range, dot: outliers. Years sharing the same letter mean not significantly different (*p* > 0.05). The grey box for 2020: the change in 2020 is significant relative to every previous year.

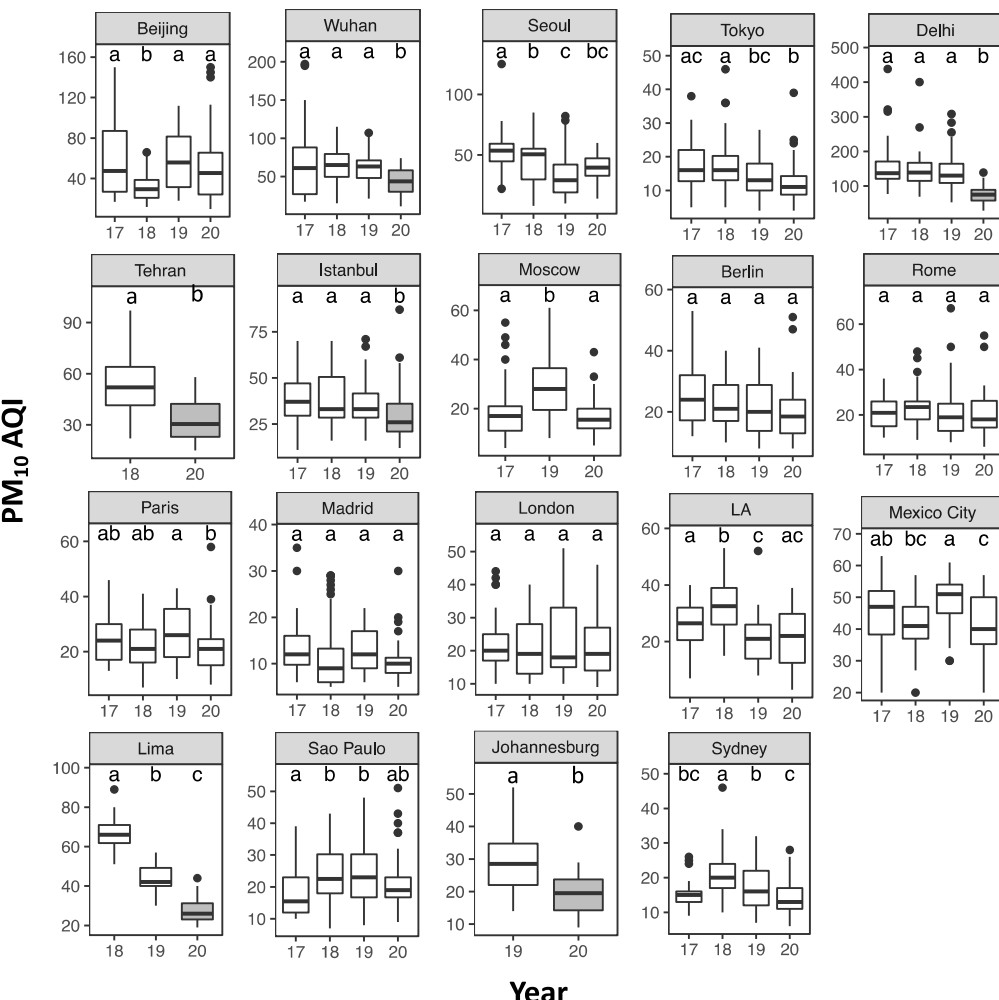

**Figure 5.** PM$_{10}$ AQI in selected worldwide cities during the lockdown period in 2020 compared to the same period of 2017, 2018 and 2019. Line within the box: the median, box: first and third quartiles, whiskers: non-outlier range, dot: outliers. Years sharing the same letter mean not significantly different ($p > 0.05$). The grey box for 2020: the change in 2020 is significant relative to every previous year.

### 3.4. Temperature and Rainfall

Temperatures in 2020 are significantly higher than that observed in 2019 for Moscow and Paris, and significantly lower in 2020 for Delhi, Tehran, NYC and Sao Paulo, with the largest temperature decrease of 2.5 °C in Sao Paulo (Figure 6). For other cities, the temperature changes in 2020 relative to 2019 were not statistically significant. Rainfall was more than two times higher in 2020 than 2019 in Beijing, Istanbul, Johannesburg, LA, Rome, Tokyo and Wuhan (Figure 6).

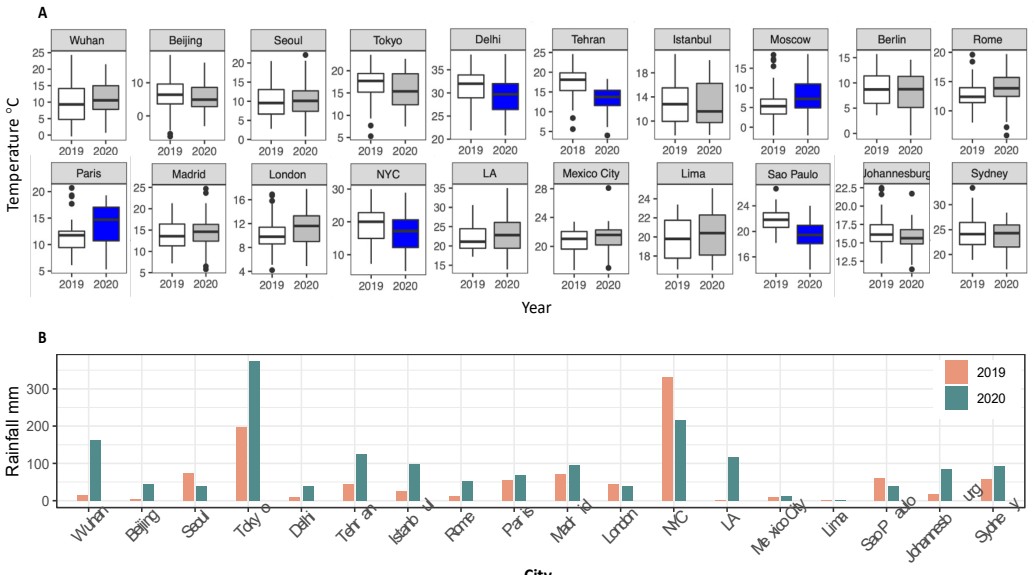

**Figure 6.** Meteorological condition changes in selected cities during lockdown period in 2020 and the same period in 2019. (**A**) Average temperature, blue boxes mean significant change of average temperature: $p < 0.05$ according to ANOVA and Tukey HSD tests. (**B**) Total rainfall, no rainfall data for Berlin and Moscow from either source.

## 4. Discussion

### 4.1. Changes in AQI of Main Pollutants

A significant impact of the COVID-19 lockdown on air quality was found in 20 major cities, including significant decreases in AQI levels for $NO_2$, $SO_2$, CO and PM, and increases of ground-level $O_3$ AQI in most cities (Table 2). The decrease of primary pollutants $NO_2$, $SO_2$, CO during the lockdown period relative to 2019 and to the average of 2017–19 for the same period are due to the reduction of emissions from anthropogenic activities. The transportation reduction is responsible for the declines of $NO_2$ and CO, due to the restriction of human mobility (e.g., automobile use decreased in all cities during lockdown [22]), and the reduced electricity consumption is responsible for the decrease of $SO_2$ due to the restrictions of industrial and commercial activities [23] (see Section 4.2 for a discussion).

For the secondary pollutant ground-level $O_3$, the AQI level depends on the $O_3$ formation rate through complex photochemical reactions, in which the three determinizing factors are sunlight, $NO_x$ and VOCs. However, the chemistry of $O_3$ formation is highly nonlinear, and the effects of precursor concentrations on $O_3$ production rate can be characterized as either $NO_x$-sensitive or VOC-sensitive [24–26]. Under a VOC-sensitive regime (low ratio of VOC/$NO_x$), an increase in $NO_x$ concentration causes a decrease of ozone with low concentrations of VOCs. On the other hand, under the $NO_x$-sensitive regime (higher ratio of VOC/$NO_x$), the reduction of $NO_x$ emissions will lead to an increase in ozone concentrations. As VOC data was not available, it's difficult to define the regime of $NO_x$-VOC-$O_3$ sensitivity. However, $O_3$ levels increased during the lockdown in most cities (Figure 3). Even in the case of Istanbul and Lima, by excluding the data from 2019, the level from 2020 was also higher than 2017 and/or 2018. Thus, ozone likely formed under a $NO_X$-sensitive regime in most of cities. Weather conditions are also another important factor, especially solar radiation or temperature.

The decline in PM reflects changes in both primary and/or secondary particles emissions and reactions. The primary PM is from natural and anthropogenic processes including road traffic within urban areas [27,28]. The two main species for secondary PM formation are sulfate and nitrate, formed in air from precursor pollutants: $SO_2$ and $NO_2$. Thus, the decline of emissions of primary pollutants $SO_2$ and $NO_2$ recorded here also indirectly reduced the formation of secondary PM.

### 4.2. Impact of Lockdown Strategy on Air Quality

Lockdown policies reduced transportation and electricity demand, reflecting restricted human mobility, industry and commercial activities [22,23]. Since the primary and secondary sources of $NO_2$ are transportation and combustion in power plants, this led to a reduction in $NO_2$ [11–13]. The percentages of decrease in $NO_2$ AQI in 20 major cities were compared to assess the impact of lockdown policy on air pollution (Figure 7), as $NO_2$ showed the most significant changes during lockdown. The car driving data [22] was also used to assess the restriction of human mobility, as transportation is the most important source of $NO_2$.

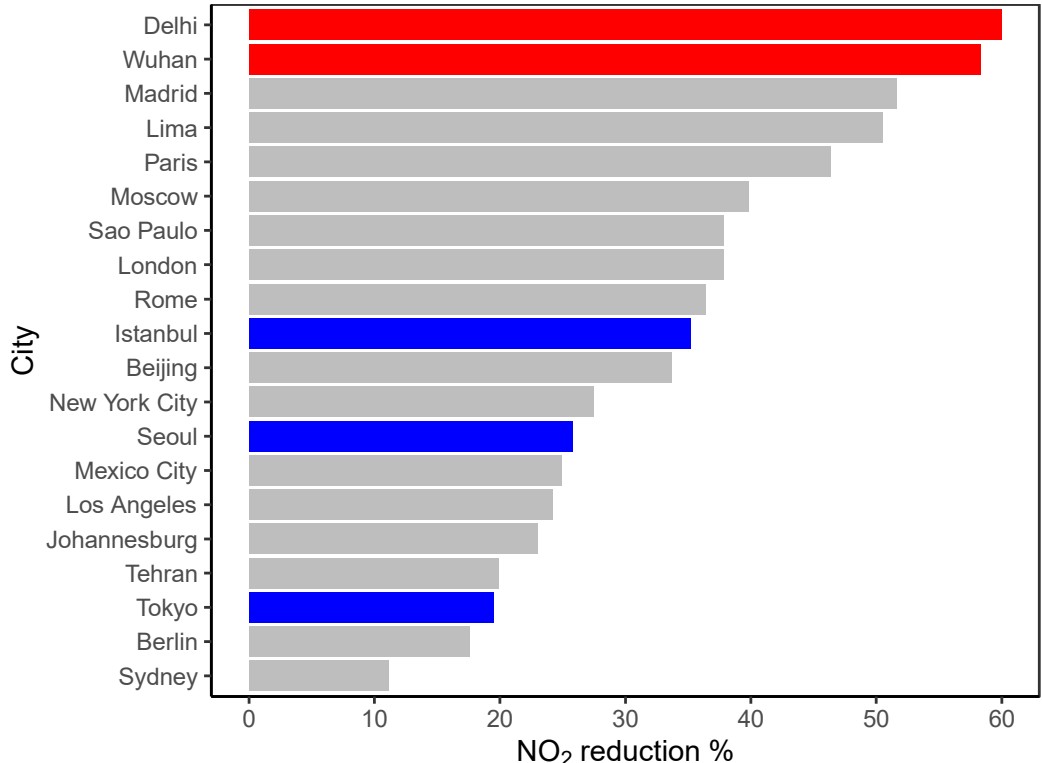

**Figure 7.** $NO_2$ reduction percentage during lockdown period in 2020 relative to 2019 for the same period in 20 major cities around the world. Red: cities with strictest lockdown policy (human mobility restriction and stopped public transportation), grey: cities with similar strict lockdown policy (full or partial lockdown, with different social distancing measures), blue: less strict lockdown policy or no lockdown.

In Wuhan, the first epicenter of COVID-19, experienced the strictest lockdown policy (Table 2). All public transport was suspended. The residents of Wuhan were not allowed to leave the city without permission from authorities, and only one person from each household was allowed to leave their house or apartment for two hours every second day for essentials [29]. A very strict lockdown policy was also applied in Delhi: public transportation was also suspended during the lockdown, and only those who worked for "essential services" could move freely. Due to the strictest lockdown policies, $NO_2$ reduced the most in Delhi and Wuhan (60.0%, 58.3%, respectively) (Figure 7). Driving in automobiles dropped about 88% in Delhi during the lockdown [22], also the highest reduction in mobility among all cities, consistent with the strictest lockdown policy and the highest reduction of $NO_2$ among all cities.

As the second epicenter of COVID-19, strict lockdown policies were implemented in hard-hit European countries, which aimed to reduce human mobility, especially in Italy, Spain, France and the UK [30–33]. For instance, people could not leave home without a permit in France, citizen mobility reduced by 68% and 79% in the Paris region, respectively, within and leaving the region [32]. In the UK,

people could only go outside to buy food, to exercise once a day, or go to work if they absolutely could not work from home [33]. The reduction of $NO_2$ in Madrid (51.6%), Rome (36.4%), London (46.4%) and Paris (37.8%) (Figure 7), agrees with the medium-strict lockdown policies implemented in those countries. Driving of automobiles declined 65–85% in these 4 European cities [22], consistent with the strictness of lockdown policies and the reduction of $NO_2$.

The Peruvian government implemented the longest lockdown (108 days, Table 2) in the world. The lockdown policy restricted citizen mobility based on gender. Only men could go out on Monday, Wednesday and Friday and only women on Tuesday, Thursday and Saturday. Otherwise, no Peruvian citizens were allowed to leave their homes, and the police and the army were deployed to enforce the lockdown [34]. Lima experienced fourth highest reduction (51.8%) of $NO_2$ relative to 2019 for the same period.

The U.S. federal government did not issue a national lockdown, but a national emergency instead. Most states issued their own stay-at-home orders. For example, stay at home orders started on March 17th in California and 22nd in New York. About 60% of the decrease in automobile use [22] was observed during stay-at-home periods in NYC and LA, which is lower than most of European cities and Delhi, meaning less strict policies applied in the U.S. than those in hard-hit European cities. The stay-at-home orders were not legally enforced. This voluntary social distancing led to a 27.5% (LA) and 24.2% (NYC) decrease of $NO_2$ AQI. This decrease was less than most of the European cities, consistent with the less strict lockdown policies in the U.S. and the lower decrease of human mobility.

The smallest declines in automobile use were observed in Sydney (50%), Berlin (50%), Seoul (45%) and Tokyo (30%) among all cities [22]. The least strict lockdown policies were implemented in Seoul and Tokyo. The Japanese government declared a "state of emergency" for seven prefectures, including Tokyo, Osaka, and Fukuoka, and implemented a Japanese version of a social-distancing policy. The government did not impose severe lockdown regulations but encouraged self-restraint on the part of the public and businesses [35]. The Korean government handled the COVID-19 crisis by applying intensive testing and contact tracing rather than enhanced social distancing, and since these measures worked to control the COVID-19 spread, a lockdown was never mandated [36]. Germany implemented a nation-wide social distancing and contact restriction on 22 March, in contrast to most other European countries, the stringency of measures differs substantially between states [37]. For example, in Berlin, people were only allowed to go out alone or with a person from same household. Australia also implemented similar lockdown policies in comparison to most European countries. All Australians were strongly advised to leave their homes only for limited essential activities and public gatherings were limited to two people. Berlin (17.6%) and Sydney (11.1%) experienced the least decreases of $NO_2$ AQI.

Among all countries, the Turkish government issued a particular policy to fight against the COVID-19 outbreak: weekend curfew for residents under the age of 20, and aged 65 and above, without full or partial lockdown [38]. However, automobile use dropped ~60%, higher than Berlin and Sydney, where statewide lockdown with human mobility restrictions were implemented. As a consequence, the decrease of $NO_2$ AQI was also one of the lowest in Istanbul, with a 19.9% decrease, also higher than those in Berlin and Sydney.

In conclusion, the decline of $NO_2$ related to the drop of automobile usage in most cases, as transportation is the main source of $NO_2$. The decrease of car use depended on the strictness of lockdown policy, especially the restrictions on human mobility and the forces that were used to control it. However, other social factors also could impact the effectiveness of lockdown policy, such as the voluntariness of citizen on social distancing [39] and the trust in government [40].

*4.3. Impact of Meteorological Conditions on Air Quality*

Year-to-year variations in meteorological conditions impacts air pollution, particularly the formation of secondary pollutants. Higher temperature/solar radiation favors the formation of ozone, and temperature correlates with ground-level ozone [5,41]. Higher temperatures can promote the

formation of ground-level $O_3$. The significant increase of $O_3$ in Paris and insignificant increase in Moscow in 2020 (Figure 3) could be partially due to the higher temperature (Figure 7). The significantly lower temperature in NYC during lockdown relative to 2019 could be responsible for the slight decrease of $O_3$. However, in Delhi and Sao Paulo, despite the significantly lower temperature during the lockdown, the $O_3$ AQI increased significantly in Sao Paulo and insignificantly in Delhi, meaning the increases of $O_3$ could be higher, if the temperature was the same as in past years.

Enhanced rainfall can wash air pollutants out of the atmosphere, especially particulate matter and water-soluble pollutants, such as $NO_2$ and $SO_2$. The decrease in primary pollutants and PM in Beijing, Istanbul, Johannesburg, LA, Rome, Tokyo and Wuhan, where rainfall was more than twice as high in 2020 than 2019, could be partially caused by this higher rainfall during the lockdown. This is especially true in Johannesburg where 4.9 times more rain fell in the 2020 lockdown period than 2019 and could explain the largest decrease of $PM_{2.5}$ (31.3%).

### 4.4. Impact of Environmental Policy on Air Quality

Prior to COVID-19, many countries imposed action plans or policies limiting emissions to the atmosphere to mitigate the impact of air pollution on public health. For instance, the Action Plan on Prevention and Control of Air Pollution in China 2013, the National Clean Air Program (NCAP) in India, the Clean Air Act in the U.S., and the National Air Pollution Control Program in the E.U. all aim to lower air pollution in their country or region. In addition to the lockdown policies and meteorological conditions, these action plans and policies could also play a role in air pollution change. This role can be assessed by observing the inter-annual variation of each pollutant prior to the occurrence of COVID.

$NO_2$ decreased significantly from 2017 to 2019 in Wuhan, Istanbul, Lima and Mexico City (Figure 2) and $SO_2$ declined significantly in Beijing, Wuhan and Mexico City (see Supplementary Figure S1). Significant decreases in $PM_{2.5}$ occurred in Delhi, Wuhan, Tokyo and Lima (Figure 4). These inter-annual declining trends could show the continuous reduction of pollutant emissions due to numerous local environmental policies implemented in these countries, especially in China and Mexico.

China and India, major developing countries in the world with the largest populations, have considerable air pollution issues especially in major cities [42,43], due to the high growth in urban population and the increased demand for energy and transportation. After the implementation of different policies to control air pollution, China experienced significant decreases (21–59%) of $PM_{2.5}$, $SO_2$ and $NO_x$, since the Action Plan on Prevention and Control of Air Pollution was implemented in 2013 [44,45]. For the decrease of air pollutants during the lockdown in 2020, pre-existing decreasing trends of air pollutants should be taken into account. The pre-existing decreasing trends of $NO_2$, $SO_2$ and $PM_{2.5}$ were reported previously in the literature and also observed in this study (Figure 2, Figure 4 and Figure S1), especially in Wuhan. In contrast, India's $SO_2$ and $NO_2$ levels increased by more than 100% and 50% from 2005 to 2015 respectively, due to the high growth of coal power plants and smelters [46]. The significant increases of $NO_2$ (Figure 2) and $SO_2$ (Figure S1) in 2019 relative to 2017 and 2018 confirm this increasing trend in Delhi. In January 2019, India launched a National Clean Air Program aimed to reduce particulate matter pollution by 20–30% by 2024 relative to 2017 levels [47]. It is too early to observe the results of this program. The increasing pollution trends of past years could cause an underestimation in the decrease of air pollution during the lockdown period.

In other major developing countries: significant pre-existing decreasing trends of $NO_2$ and $SO_2$ were also found in Mexico City during 2017–2019. Since the 1990s, the Mexican government developed and implemented successive air pollution programs that combined regulatory actions with technological changes that resulted in significant improvement to air quality. $PM_{2.5}$ (60%) $NO_2$ (40%) and $SO_2$ (90%) decreased dramatically since 1990 to 2018 (Molina et al., 2019). This decreasing trend could partially be responsible for the lower concentrations of $NO_2$ and $SO_2$ during 2020 lockdown in Mexico City. In Turkey, according to the regulations, every Provincial Directorate of Environment and Urban Planning has to prepare a clean air plan. The concentration of $PM_{10}$ and $SO_2$ has decreased by 50% and 98% respectively since 1990s to 2014 [48], due to numerous measures included in a clean air

action plan. Similar to Mexico City, the decrease of $NO_2$ could be underestimated in considering the pre-existing decreasing trend of $NO_2$ in Istanbul. However, a significant increasing trend of $SO_2$ was observed from 2017 to 2020 (Figure S1), signifying the increase of $SO_2$ during 2020 lockdown was a continuous trend, but not specifically caused by the COVID-19 lockdown.

In contrast with Asian and South American countries, air pollution concentrations in European countries and the United States remain stable and at a relatively lower level compared to most Asian and South American countries. This is due to earlier urbanization and implementation of air pollution action plans. In the E.U. countries, the Convention on Long-Range Transboundary Air Pollution (LRTAP) was signed in 1979, aiming to mitigate the air pollution transmitted over long distances by reducing emissions and pollution prevention [49]. Since 1980, numerous directives on the limitations of air pollution concentrations have been implemented [50], and the air quality has been improved in many European countries. For instance, since 2000 in the Paris region the $PM_{2.5}$ concentration is lower than the World Health Organization suggested limit (25 $\mu g/m^3$). $NO_2$ and $SO_2$ concentrations also remain stable over the past 4 years [12]. The U.S. implemented the Clean Air Act (CAA) in 1970, which dramatically improved air quality in the U.S. nationally, concentrations of air pollutants in 2019 dropped significantly compared to 2000: 92% ($SO_2$), 62% ($NO_2$), and $PM_{2.5}$ (43%) [51]. However, in the last 4–5 years, the national annual average of air pollutants is stable. Due to the relatively stable concentrations of air pollutants, especially $NO_2$, $SO_2$ and $PM_{2.5}$ in those countries, the impact of environmental policies on short-term air quality should be too low to observe.

## 5. Conclusions

Significant decreases in the AQI of $NO_2$, $SO_2$, CO, $PM_{2.5}$ and $PM_{10}$ were observed during the lockdown period in most of 20 megacities in the world relative to 2019 and to the 2017–2019 average for the same period. For the primary pollutants: $SO_2$, $NO_2$ and CO, the significant decreases were directly due to the reduction of emissions caused by lockdown, as citizen mobility was restricted. The difference of $NO_2$ reduction between cities was mainly due to the various lockdown policies, and Wuhan and Delhi exhibited the highest decrease of $NO_2$ due to the strictest lockdown policies. For the secondary pollutants, $O_3$ increased in most of cities, due to photochemical reactions promoting ozone formation under a potential VOC-sensitive regime. $PM_{2.5}$ and $PM_{10}$ decreased in 19 and 17 of all cities respectively, but the decrease was less than its precursor gases, especially $NO_2$, as the sources of PM are complex.

Meteorological variability also plays a role in air pollutant concentration: significantly higher rainfall during the lockdown period in Johannesburg could explain the largest decline of $PM_{2.5}$, and the lower temperature and higher rainfall in Istanbul and Tokyo could explain the exceptional decrease of ozone. In addition, environmental policy regulations, especially in Asian cities, such as Beijing, Wuhan, Seoul and Tokyo, reduced pollutant emissions leading to decreasing $NO_2$, $SO_2$ and $PM_{2.5}$ concentrations through the past three years prior to the lockdown. Globally, despite the non-negligible impacts of meteorological variability and preceding environmental policy, lockdown explains the large reductions in air pollution.

**Supplementary Materials:** The following are available online at http://www.mdpi.com/2073-4433/11/11/1189/s1, Figure S1: SO2 AQI in selected worldwide cities during the lockdown period in 2020 compared to the same period of 2017, 2018 and 2019. Figure S2: CO AQI in selected worldwide cities during the lockdown period in 2020 compared to the same period of 2017, 2018 and 2019. Table S1: Name and number of stations for Meteorological data for 20 selected major cities.

**Author Contributions:** All authors made significant and extensive contributions to the work presented in this manuscript. F.F., K.L.P.-R. and B.W. conceived the idea, all authors designed and discussed data. F.F. conducted the data analysis and wrote the manuscript with guidance from K.L.P.-R. and B.W., all co-authors contributed critical revisions. All authors have read and agreed to the published version of the manuscript.

**Funding:** This work was supported by Envirolab Asia and the Henry Luce Foundation.

**Conflicts of Interest:** The authors declare no conflict of interest.

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
