# Peer review of "Impact of the COVID-19 Pandemic Lockdown on Air Pollution in 20 Major Cities around the World"

_atmosphere, doi:10.3390/atmos11111189_

Round 1
Reviewer 1 Report
Fu et al. presented a comprehensive comparison of the AQIs in 20 cities all over the world during the COVID-19 lockdown period with those in previous years. They found that air pollutants due to anthropogenic emissions were significantly reduced during the lockdown period. Meteorological conditions also played a role in determining air quality. The data analysis is overall very well presented and the manuscript is well organized and easy to follow, thus should be of interest to people not in this filed. I recommend for publication only after a minor change:
Does the AQI data just include min, max, median, and standard deviation of each day instead of hourly data? Since all the data is presented in statistical mode, the temporal information is lost, which could be very important and intuitive, e.g., to show the impact of the meteorological condition on air quality. It will be better to include this data in either the manuscript or supplementary material (and mark rainy days).
Author Response
Review comments: Does the AQI data just include min, max, median, and standard deviation of each day instead of hourly data?
Answer: This study used median AQI of each day, this information has been added in the line 93.
Review comments: Since all the data is presented in statistical mode, the temporal information is lost, which could be very important and intuitive, e.g., to show the impact of the meteorological condition on air quality. It will be better to include this data in either the manuscript or supplementary material (and mark rainy days).
It should be interesting to assess the temporal variation of meteorological data and air pollution. We would probably find some correlation between them. However, this paper focused on the interannual comparison, only the average of the AQI of pollutants during lockdown in 2020 was compared to previous years.
Reviewer 2 Report
The specific work is dealing with the most pressing issue of our days, the COVID-19. To be more specific, it discusses the impact of the different lockdown strategies of 20 different cities to their configured air quality status. In other words, it investigates the impact of the different source apportioning to the concentration levels of five important air pollutants: NO2, SO2, CO, PM2.5, PM10. All the new data sets, especially those which are strongly associated with the everyday life are welcome to the literature.
To this end, in my opinion, the specific work can be accepted fro publication after minor changes.
A general comment: please write on the third person (line 10, 69, 74, 97, 101, 109, 116, 118, 121, 151, 157, 204, 220, 222, 240, 242, 328, 341, 351, 361, 381)
Introduction
line 31:.... are routinely measured in many countries.
line 34-46: please make clear that most air pollutants are both primary and secondary originated and then discuss their twin identity.
Materials and Methods
line 87: .. the data set provides a statistical summary....
Author Response
Reviewer comments: A general comment: please write on the third person (line 10, 69, 74, 97, 101, 109, 116, 118, 121, 151, 157, 204, 220, 222, 240, 242, 328, 341, 351, 361, 381)
Answer: We replaced all “we” by passive phrases.
Reviewer comments: line 31:.... are routinely measured in many countries.
Answer: Corrected according to the suggestion.
Reviewer comments: line 34-46: please make clear that most air pollutants are both primary and secondary originated and then discuss their twin identity.
Answer: Corrected according to the suggestion in line 37-38.
Reviewer comments: line 87: .. the data set provides a statistical summary....
Answer: Corrected according to the suggestion.
Reviewer 3 Report
The manuscript addresses a very actual and interesting paper. It is well organized and supported with adequate Figures and Tables. Despite being mentioned in different sections of the manuscript, no supplementary material was found nor in the submitted manuscript nor online at the MDPI system. Please also consider some suggestions to improve the manuscript.
Figure 1:
- i) please check the position of the localizations 11-13 on the map.
- ii) in the caption authors mention "Twenty major cities impacted by COVID-19" and so, something to support the expression "major cities impacted" should be added in the introduction. Also, which was the criteria to select the megacities considered in the study?
Table 1:
- i) From my point of view the Lockdown Policy could be improved. As it is, it is not clear the main policies adopted. Maybe it would help if the lockdown policy column was replaced by 2 columns to indicate (yes/no) the occurrence of total/partial (...) lockdown. And then if needed give some extra information to total and partial lockdown in footnotes of the table or in an additional column (e.g., other local actions).
Table 2:
“+” means increase of AQI but the values presented in the table represent percentage of decrease; it gets confuse. Do values represent the decrease unless indicated as "+"?
Authors should try to clarify to avoid miss understandings.
Figure S3, information not found. If Figure S3 is similar to Figures 2-4, then it could be moved to the manuscript. If the number of figures and tables are a limitation of the journal, then I suggest moving Figure S3 to the manuscript and the Figure 5 to supplementary material.
Figure 5: A different color could be used for the 2020 values of rainfall to become different from the mean increase/decrease marked in the temperature values.
Line 156159: By looking to Table 2 it seems that the maximum decrease occurred in Beijing (+92.4) also relative to 2019. It is something related with the +? If so, please adjust the table to become clearer.
Lines 177-179: Looking to Figure 4, levels of PM2.5 do not seem to decrease in 2020 relatively to 2019 and/or 2017-2019 for some countries (e.g., Madrid, London). Please check and if necessary, revise. Also, the sentence has an additional parenthesis.
Line 148: Table 2 only has 6 red bold values, thus meaning that SO2 decreased significantly in 6 cities. Please check and revise.
English should be revised. Please consider some suggestions:
Lines 23: through damage in the lungs (...).
Lines 27-28: (...) are all associated with the development and/or aggravation of respiratory diseases (...)
Lines 29-30: the sentence should be rephrased since usually the airborne levels of CO can aggravated some diseases but should not cause the dead; dead may occur in some specific situations.
Line 261; Paris, repetition.
Author Response
Figure 1:
i) please check the position of the localizations 11-13 on the map.
Answer: Positions of London and Madrid are switched.
ii) in the caption authors mention "Twenty major cities impacted by COVID-19" and so, something to support the expression "major cities impacted" should be added in the introduction. Also, which was the criteria to select the megacities considered in the study?
Answer: Modified accordingly in lines 61, and 68-70.
Table 1:
i) From my point of view the Lockdown Policy could be improved. As it is, it is not clear the main policies adopted. Maybe it would help if the lockdown policy column was replaced by 2 columns to indicate (yes/no) the occurrence of total/partial (...) lockdown. And then if needed give some extra information to total and partial lockdown in footnotes of the table or in an additional column (e.g., other local actions).
Answer: Modified by following the suggestions
Table 2:
“+” means increase of AQI but the values presented in the table represent percentage of decrease; it gets confuse. Do values represent the decrease unless indicated as "+"? Authors should try to clarify to avoid miss understandings.
Answer: Table 2 has been modified, “-“ and “+” were defined as decrease and increase of AQI respectively.
Figure S3, information not found. If Figure S3 is similar to Figures 2-4, then it could be moved to the manuscript. If the number of figures and tables are a limitation of the journal, then I suggest moving Figure S3 to the manuscript and the Figure 5 to supplementary material.
Answer: Figure S3 (PM10) has been added as Figure 5 in the manuscript.
Figure 5: A different color could be used for the 2020 values of rainfall to become different from the mean increase/decrease marked in the temperature values.
Answer: Figure 5 (now Figure 6) has been modified, other colors were used for rainfall.
Line 156159: By looking to Table 2 it seems that the maximum decrease occurred in Beijing (+92.4) also relative to 2019. It is something related with the +? If so, please adjust the table to become clearer.
Answer: Table 2 has been modified, “-“ and “+” were defined as decrease and increase of AQI respectively.
CO had a very low level in Beijing in 2019, that’s why there was an very high increase of CO in Beijing during the lockdown period relative to 2019 (Figure S2 added in supplementary materials). This abnormal increase was also discussed in lines 161-162.
Lines 177-179: Looking to Figure 4, levels of PM2.5 do not seem to decrease in 2020 relatively to 2019 and/or 2017-2019 for some countries (e.g., Madrid, London). Please check and if necessary, revise. Also, the sentence has an additional parenthesis.
Answer: The additional parenthesis was deleted, in Madrid and London, the decreases of NO2 AQI were not significant, the medians were similar to that of 2019. However, in the Table 2, the percentage change of NO2 was the Mean, so it can be a little bit different to Median.
Line 148: Table 2 only has 6 red bold values, thus meaning that SO2 decreased significantly in 6 cities. Please check and revise.
Answer: The number of cities with significant decrease of SO2 relative to last 3 years was 6, it’s been modified in the line 148 (now 149).
English should be revised. Please consider some suggestions:
Lines 23: through damage in the lungs (...).
Answer: Modified according to the suggestion.
Lines 27-28: (...) are all associated with the development and/or aggravation of respiratory diseases (...)
Answer: Modified according to the suggestion.
Lines 29-30: the sentence should be rephrased since usually the airborne levels of CO can aggravated some diseases but should not cause the dead; dead may occur in some specific situations.
Answer; According to the paper cited (Raub et al., 2000), at low concentrations Co can cause subtle cardiovascular and neurobehavioral effects at low concentrations to unconsciousness and death after acute or chronic exposure to higher concentrations. To avoid this misunderstanding, I only used the low concentrations, as the high concentration is not really related to the outdoor air pollution. (see correction in lines 30-31)
Line 261; Paris, repetition.
Answer: Paris in parenthesis has been deleted.